# Sensitivity and specificity of commercially available rapid diagnostic tests for viral hepatitis B and C screening in serum samples

**Ganbolor Jargalsaikhan**[1,2☯], **Miriam Eichner**[1,2☯], **Delgerbat Boldbaatar**[1,2], **Purevjargal Bat-Ulzii**[1,2], **Oyungerel Lkhagva-Ochir**[1,2], **Odgerel Oidovsambuu**[1,2,3], **Bekhbold Dashtseren**[1,2], **Erdenebayar Namjil**[4], **Zulkhuu Genden**[1,2], **Dahgwahdorj Yagaanbuyant**[1,2,5], **Alimaa Tuya**[4], **Naran Gurjav**[4], **Altankhuu Mordorj**[1,2], **Andreas Bungert**[1,2]*, **Naranbaatar Dashdorj**[1,2‡], **Naranjargal Dashdorj**[1,2‡]

**1** Liver Center, Sukhbaatar District, Ulaanbaatar, Mongolia, **2** Onom Foundation, Khan-Uul District, Ulaanbaatar, Mongolia, **3** National University of Mongolia, Sukhbaatar District, Ulaanbaatar, Mongolia, **4** National Center of Transfusion Medicine, Sukhbaatar District, Ulaanbaatar, Mongolia, **5** Mongolian National University of Medical Sciences, Ulaanbaatar, Mongolia

☯ These authors contributed equally to this work.
‡ These authors also contributed equally to this work.
* Andreas.Bungert@onomfoundation.org

**Data Availability Statement:** The file containing the raw data is available from figshare.com (https://doi.org/10.6084/m9.figshare.9724157).

## Abstract

Early diagnosis of chronic hepatitis B virus (HBV) and hepatitis C virus (HCV) infections is pivotal for optimal disease management. Sensitivity and specificity of 19 rapid diagnostic test (RDT) kits by different manufacturers (ABON, CTK Biotech, Cypress Diagnostics, Green Gross, Human Diagnostic, Humasis, InTec, OraSure, SD Bioline, Wondfo) were assessed on serum samples of 270 Mongolians (90 seropositive for hepatitis B surface antigen (HBsAg), 90 seropositive for hepatitis C antibody (HCV-Ab), 90 healthy subjects). All tested RDTs for detection of HBsAg performed with average sensitivities and specificities of 100% and 99%, respectively. Albeit, overall sensitivity and specificity of RDTs for detection of HCV-Ab was somewhat lower compared to that of HBsAg RDTs (average sensitivity 98.9%, average specificity 96.7%). Specificity of RDTs for detection of HCV-Ab was dramatically lower among HBsAg positive individuals, who were 10.2 times more likely to show false positive test results. The results of our prospective study demonstrate that inexpensive, easy to handle RDTs are a promising tool in effective HBV- and HCV-screening especially in resource-limited settings.

## Introduction

With about 1.4 million annual deaths, viral hepatitis is a major problem in global health [1,2]. Most deaths are attributable to chronic hepatitis B virus (HBV) and hepatitis C virus (HCV) infections and their long-term complications–liver cirrhosis and hepatocellular carcinoma. Early diagnosis and linkage to care is pivotal to prevent these complications. However, as early stages of HBV and HCV infection are often asymptomatic, few people are diagnosed early. In

**Funding:** The author(s) received no specific funding for this work.

**Competing interests:** The authors have declared that no competing interests exist.

2015, about 9% of 257 million individuals living with chronic HBV and 20% of the 71 million with chronic HCV infection were aware of their disease status globally [1].

For reducing the global burden of hepatitis, identifying those who are infected is crucial. This is especially true for HCV, since in recent years highly effective direct acting antiagents (DAAs) have become available as a reliable cure against the disease [3]. Similarly, for HBV, antiviral treatment using nucleoside analogues are not only effective in inhibiting the progress of the disease, but also in preventing the transmission of the disease [4,5]. However, for diagnostics, costs, local access and ease of use are important to reach a large population.

Especially in low- to middle-income countries like Mongolia with high HBV and HCV prevalence [6], reliable rapid diagnostic tests (RTD) represent a promising alternative to standard testing methods like enzyme linked immunosorbent assay (ELISA) for initial screening. RDTs are cheaper, quicker to perform and require less skill and instrumentation. Performance of RDTs for hepatitis B surface antigen (HBsAg) and hepatitis C antibody (HCV-Ab) detection by different manufacturers has been reported to vary [7–10]. Therefore, the performance of RDTs should be carefully assessed before use in clinical practice.

The aim of this study was to evaluate the diagnostic performance of commercially available RDTs for HBsAg and HCV-Ab detection.

## Materials and methods

### Patients

270 participants were prospectively recruited: 90 HBsAg positive, 90 with detectable HCV viral load (HCV-RNA positive), 90 healthy controls. The sample size of 90 participants per group is a compromise between costs and statistic accuracy, to which the reliability of RDTs can be determined.

HBsAg or HCV-RNA positive participants were randomly selected from screening registry of the Liver Center, Ulaanbaatar, Mongolia. Inclusion criteria: ≥ 18 years, positive tests for HBsAg or HCV-RNA within the year prior to the study. Patients with dual infection were excluded.

Healthy controls were randomly selected among blood donors at the National Center of Transfusion Medicine, Ulaanbaatar, Mongolia. Inclusion criterium: Three or more blood donations, ensuring that these participants were confirmed negative for HBsAg, HCV-Ab and other common chronic infectious diseases multiple times.

Ethical approval was obtained from the Ethics Committee of the Ministry of Health, Mongolia (approval number 14-12/1A). Each individual gave written informed consent prior to participation.

### Performance of study and laboratory measurements

Participants were asked to provide a blood sample at the Liver Center or the National Center of Transfusion Medicine, respectively, between April and July 2015. From each individual, two samples of 5 ml venous blood were collected into vials containing clotting agent (Greetmed, Vacuum Tube Clot Activator). All further processing and testing of blood samples was performed at the Liver Center. Within 4 hours after blood drawn, serum was separated by centrifugation (1110 × g; 5 min; room temperature) and stored at -80˚ C until reference or index testing.

As reference, HBsAg and HCV-Ab status of all serum samples was determined by ELISA (DiaPro HBsAg and HCV-Ab 3$^{rd}$ generation ELISA) following manufacturer's instructions. All samples were further checked by fully automated quantitative RT-PCR (Abbot, m2000) for quantitation of HBV-DNA and HCV-RNA according to manufacturer's instructions. For cost

reasons, a total of 90 seronegative samples of healthy controls was analyzed by RT-PCR as 3 pooled samples (3×30). This decreases sensitivity by a factor of 30.

We assessed 9 RDTs for HBsAg, 10 for HCV-Ab detection from different manufacturers (Table 1).

Most RDTs were purchased by local distribution partners, which were not aware that tests were to be used for quality assessment. Therefore, the shown prices represent end-customer prices for low quantities in early 2015 in Mongolia. OraSure and InTec were donated by manufacturers.

All RDTs–except those from InTec–were performed simultaneously according to manufacturer's instructions. Fresh aliquots of serum were defrosted at room temperature and samples applied to test wells by pipetting. This was done for all 270 samples by end of July 2015. Final test outcome was subject to visual inspection by the researcher. Kits from InTec were only later (in February 2017) included upon request of the manufacturer. In all cases, the researcher was aware of the sample type and test he or she was assessing.

## Data analysis, statistics

Sensitivity and specificity were determined for every test using ELISA (HBsAg and HCV-Ab) results as reference. 95% confidence intervals (95% CI) for sensitivity and specificity were calculated using Wilson score method without continuity correction (6). Positive- and negative likelihood (LR+ and LR-) ratios were calculated based on the values for sensitivity and specificity.

To put the results in the context of hepatitis screening activities in Mongolia, we assumed, as previously reported, a prevalence for HBsAg of 11.0% and for HCV-Ab of 8.5% among Mongolian adults (6). These values were used to calculate positive- and negative predictive values (PPV and NPV) and diagnostic accuracy (DA) [11].

**Table 1. Overview of tests kits and manufacturer.** Prices are given as payed when purchased in 2015 in small quantities by local distributors in Mongolia. All manufacturers, except OraSure, offer RDTs for HBsAg and for HCV-Ab.

| Manufacturer | Distributor in Mongolia | Product | Lot Number | Price per test (USD) |
|---|---|---|---|---|
| Abon Biopharm, Hangzhou, China | Lifetronik, LLC | One Step HBsAg rapid test | BSG4120041 | 0.75 |
| | | One step HCV antibody rapid test | F0805K3B00 | 0.75 |
| CTK Biotech, San Diego, USA | Monos Group & IldenGun LLC | Onsite HBsAg combo rapid test | F0321L6D00 | 0.65 |
| | | Onsite HCV Ab plus combo rapid test | HCV4120057 | 0.9 |
| Cypress Diagnostics, Belgium | MonBioPharm, LLC | HBsAg Dipstick test | B20140520 | 0.75 |
| | | Anti-HCV dipstick test | B201503056 | 0.9 |
| Green Cross Life Sciences Corp, Korea | MEIC, LLC & MongolPharm, LLC | Genedia HBsAg | 346A1501 | 0.45 |
| | | Genedia HCV rapid LF | 148A0034 | 0.6 |
| Human, Wiesbaden, Germany | MonoLab, LLC | Hexagon HBsAg test | 58003 | 0.8 |
| | | Hexagon HCV test | 58072 | 1 |
| Humasis, Gyeonggi-do, Korea | MonBioPharm, LLC | Humasis HBsAg card | BSGC4003 | 0.75 |
| | | Humasis HCV card | CBCC5002 | 0.9 |
| InTec Products, Xiamen, China | IldenGun LLC | One step HBsAg test card | 2016060935 | donated |
| | | One step HCV test card | 6642962 | donated |
| OraSure technologies, Bethlehem, PA, USA | No official distributor in Mongolia | n/a | n/a | |
| | | OraQuick HCV rapid antibody test | 6642962 | donated |
| SD Standart Diagnostics Ltd, Kyonggi-do, Korea | MedImpex, LLC | SD Bioline HBsAg test | 01AD14001 | 0.7 |
| | | SD Bioline HCV test | 02BD14009 | 0.78 |
| Wondfo Biotech, Guangzhou, China | Lifetronik, LLC | One step Hepatitis B Virus | W00340705W | 0.6 |
| | | One step Hepatitis C Virus | W00540702W | 0.8 |

Additional statistical analyses were conducted using Fisher´s Exact Test.

## Results

### Characteristics of study population

270 participants in 3 groups of 90 individuals were included in this study. 37% were of male gender. Mean age was 43.2 (18–75) years. None of the patients had any history of treatment with DAAs against HCV or nucleoside analogues against HBV since these drugs only gained widespread availability on the Mongolian market in late 2015.

ELISA confirmed serostatus of participants recruited as HBsAg positives and healthy controls. All recruited HCV-RNA positive individuals were found to be HCV-Ab seropositive in ELISA. For one participant from the HBsAg positive group, the respective HBV-DNA level was zero. For two participants from the HCV-Ab positive group, HCV-RNA results were positive, but below detection limit of the assay (<20 iU/ml). Further details see Table 2.

### Results of HBsAg and HCV-Ab RDTs in serum testing

In total, 5400 RDTs were carried out within this study, 2700 for HBsAg and 2700 for HCV-Ab. None of the RDTs gave any invalid test result, which means, that the control line appeared on each test kit.

Among all RDTs for HBsAg detection, there were no false negative (FN) results and 6 out of 9 kits had no false positive results (FP) (Table 3). Only kits from CTK, Human and Cypress Diagnostics had some FN results (specificities were 98.3% (95.2–99.4%), 97.2% (93.7–98.8%) and 95.6% (91.5–97.7%), respectively). Average sensitivity for HBsAg kits was 100% (95.9–100%), average specificity 99% (96.4–99.6%). HCV-Ab serostatus had no effect on number of false positive test results with HBsAg kits (p>0.05) with 7 out of 16 FP tests coming from the HCV-Ab positive group, odds ratio 0.78 (0.29–2.09).

For HCV-Ab detection, the OraSure test showed 100% in sensitivity and specificity (Table 4). Average sensitivity for all HCV-Ab tests was 98.9% (94.2–99.6%) and the average specificity was 96.4% (92.9–98.1%).

Among 64 FP results for HCV-Ab detection, 91% occurred among HBsAg positives, odds ratio 10.2 (4.4–23.8 95%CI). The average specificity for HCV-Ab among HBsAg negatives was 99.3% compared to a specificity among HBsAg positives of 93.6% (Table 5). Specifically RDTs for HCV-Ab detection from 3 manufacturers (ABON, CTK, Humasis) were significantly more likely to give false positive results in HBsAg positives than healthy individuals (p<0.05) and further 4 manufacturers produced FP results exclusively among HBsAg positives (Cypress, Green Gross, Intec, SD-Bioline).

**Table 2. Demographic and virological characteristics of the study population including the control group, HBsAg seropositive and HCV-RNA positive group.**

|  | Control | HBsAg | HCV-RNA positive |
|---|---|---|---|
|  | n = 90 | n = 90 | n = 90 |
| Age, mean (range) | 33.4 (18–58) | 43.9 (18–75) | 52.2 (21–75) |
| male (%) | 18 (20) | 42 (47) | 41 (46) |
| HBsAg ELISA positive, n (%) | 0 | 90 (100) | 0 |
| HBV DNA level, log IU/ml, >1, n (log(U/ml) mean±SD) | 0 | 89 (2.5±1.6) | 0 |
| HBV DNA level, > 20,000 IU/ml, n (%) | 0 | 10 (11) | 0 |
| HCV-Ab ELISA positive, n (%) | 0 | 0 | 90 (100) |
| HCV RNA level, log IU/ml, >1.4, n (log(iU/ml) mean±SD) | 0 | 0 | 88 (5.5±1.3) |
| HCV RNA level, > 800,000 IU/ml, n (%) | 0 | 0 | 43 (48) |

**Table 3. RDTs for detection of HBsAg in blood serum.** Each test was applied to 90 condition positive (P) and 180 condition negative (N) participants. Results of HBsAg ELISA served as reference standard. For the average, the 95% CI was averaged across all tests. For PPV, NPV and DA, an HBsAg prevalence of 11.0% was assumed.

| Company; Product | ABON Biopharm; One Step HBsAg | CTK; OnSite HBsAg Combo Rapid | Cypress Diagnostics; HBsAg Dipstick | Green Cross; Genedia HBsAg Rapid | Human Diagnostic; Hexagon HBsAg | Humasis; HBsAg Card | InTec; HBsAg Rapid | SD-Bioline; One Step HBsAg | Wondfo; One Step HBsAg | Sum/Average |
|---|---|---|---|---|---|---|---|---|---|---|
| **FN/P (%)** | 0/90 (0) | 0/90 (0) | 0/90 (0) | 0/90 (0) | 0/90 (0) | 0/90 (0) | 0/90 (0) | 0/90 (0) | 0/90 (0) | **0/810 (0)** |
| **FP/N (%)** | 0/180 (0) | 3/180 (1.7%) | 8/180 (4.4%) | 0/180 (0) | 5/180 (2.8%) | 0/180 (0) | 0/180 (0) | 0/180 (0) | 0/180 (0) | **16/1620 (1.0%)** |
| **Sensitivity (95% CI)** | 100% (95.9–100%) | 100% (95.9–100%) | 100% (95.9–100%) | 100% (95.9–100%) | 100% (95.9–100%) | 100% (95.9–100%) | 100% (95.9–100%) | 100% (95.9–100%) | 100% (95.9–100%) | **100% (95.9–100%)** |
| **Specificity (95% CI)** | 100% (97.9–100%) | 98.3% (95.2–99.4%) | 95.6% (91.5–97.7%) | 100% (97.9–100%) | 97.2% (93.7–98.8%) | 100% (97.9–100%) | 100% (97.9–100%) | 100% (97.9–100%) | 100% (97.9–100%) | **99% (96.5–99.6%)** |
| **LR+ (95% CI)** | ∞ (46–∞) | 60 (20–175) | 23 (11–44) | ∞ (46–∞) | 36 (15–84) | ∞ (46–∞) | ∞ (46–∞) | ∞ (46–∞) | ∞ (46–∞) | **101.3 (27–222)** |
| **LR- (95% CI)** | 0 (0–4.1%) | 0 (0–4.3%) | 0 (0–4.5%) | 0 (0–4.1%) | 0 (0–4.3%) | 0 (0–4.1%) | 0 (0–4.1%) | 0 (0–4.1%) | 0 (0–4.1%) | **0 (0–4.2%)** |
| **PPV** | 100% (85–100%) | 88.1% (71–96%) | 73.6% (58–84%) | 100% (85–100%) | 81.6% (65–91%) | 100% (85–100%) | 100% (85–100%) | 100% (85–100%) | 100% (85–100%) | **92.6% (77–97%)** |
| **NPV** | 100% (99.5–100%) | 100% (99.5–100%) | 100% (99.5–100%) | 100% (99.5–100%) | 100% (99.5–100%) | 100% (99.5–100%) | 100% (99.5–100%) | 100% (99.5–100%) | 100% (99.5–100%) | **100% (99.5–100%)** |
| **DA** | 100% (97.7–100%) | 98.5% (93.5–99.5%) | 96.0% (92.0–98.0%) | 100% (97.7–100%) | 97.5% (93.9–98.9%) | 100% (97.7–100%) | 100% (97.7–100%) | 100% (97.7–100%) | 100% (97.7–100%) | **99.1% (96.4–99.6%)** |

Overall, sensitivities and specificities for HBsAg kits were better compared to HCV with with 0 FN and 16 FP among in total 2430 test for HBsAg compared to 10 FN and 64 FP among 2700 for HCV-Ab. However, the difference in specificity is not significant (p<0.05), when excluding HBsAg for detection of HCV-Ab.

**Table 4. Results of RDTs for detection of HCV-Ab in blood serum.** For PPV, NPV and DA, an HCV-Ab prevalence of 8.5% was assumed.

| Company; Product | ABON Biopharm; One Step HCV-Ab | CTK; OnSite HCV Ab Plus; Combo Rapid | Cypress Diagnostics; HCV-Ab Dipstick | Green Cross; Genedia; HCV Rapid LF | Human Diagnostic; Hexagon HCV | Humasis; HCV Card | InTec; HCV-Ab Rapid | OraSure; OraQuick HCV | SD Bioline; HCV | Wondfo; One Step HCV | All or Average |
|---|---|---|---|---|---|---|---|---|---|---|---|
| **FN/P** | 0/90 (0) | 0/90 (0) | 3/90 (3.3%) | 3/90 (3.3%) | 1/90 (1.1%) | 0/90 (0) | 0/90 (0) | 0/90 (0) | 3/90 (3.3%) | 0/90 (0) | **10/900 (1.1%)** |
| **FP/N** | 25/180 (13.9%) | 6/180 (3.3%) | 4/180 (2.2%) | 2/180 (2.2%) | 4/180 (2.2%) | 11/180 (6.1%) | 4/180 (2.2%) | 0/180 (0) | 2/180 (1.1%) | 6/180 (2.2%) | **64/1800 (3.6%)** |
| **Sensitivity (95% CI)** | 100% (95.9–100%) | 100% (95.9–100%) | 96.7% (90.7–98.9%) | 96.7% (90.7–98.9%) | 98.9% (94–99.8%) | 100% (95.9–100%) | 100% (95.9–100%) | 100% (95.9–100%) | 96.7% (90.7–98.9%) | 100% (95.9–100%) | **98.9% (94.2–99.6%)** |
| **Specificity (95% CI)** | 86.1% (80.3–90.4%) | 96.7% (92.9–98.5%) | 97.8% (94.5–99.1%) | 98.9% (96.1–99.7%) | 97.8% (94.5–99.1%) | 93.9% (89.4–96.6%) | 97.8% (94.5–99.1%) | 100% (97.9–100%) | 98.9% (96.1–99.7%) | 96.7% (92.9–98.5%) | **96.4% (92.9–98.1%)** |
| **LR+ (95% CI)** | 7.2 (4–10) | 30 (14–65). | 43.5 (16–113) | 87 (23–322) | 44.5 (17–114) | 16.4 (9.1–29) | 45 (17–115) | ∞ (46–∞) | 87 (23–322) | 30 (14–65) | **27.8 (13–1)** |
| **LR- (95% CI)** | 0 (0–5%) | 0 (0–4.4%) | 3.4% (1.2–9.9%) | 3.4% (1.1–10%) | 1.1% (0.2–6.3%) | 0 (0–4.3%) | 0 (0–4.1%) | 0 (0–4.1%) | 3.4% (1.1–9.7%) | 0 (0–4.4%) | **1.2% (0.4–6.3%)** |
| **PPV (95% CI)** | 40.1% (31–49%) | 73.6% (56–86%) | 80.2% (60–91%) | 89.0% (68–97%) | 80.5% (61–91%) | 60.3% (46–73%) | 80.7% (62–91%) | 100.0% (81–100%) | 89.0% (68–97%) | 73.6% (56–85%) | **72.1% (55–83%)** |
| **NPV (95% CI)** | 100.0% (99.5–100%) | 100.0% (99.6–100%) | 99.7% (99.1–99.9%) | 99.7% (99.1–99.9%) | 99.9% (99.4–100%) | 100.0% (99.6–100%) | 100.0% (99.6–100%) | 100.0% (99.6–100%) | 99.7% (99.1–99.9%) | 100.0% (99.6–100%) | **99.9% (99.4–100%)** |
| **DA (95% CI)** | 87.3% (81.7–91.2%) | 97.0% (93.2–98.6%) | 97.7% (94.1–99.1%) | 98.7% (95.6–99.6%) | 97.9% (94.4–99.2%) | 94.4% (90.0–96.8%) | 98.0% (94.6–99.2%) | 100.0% (97.8–100%) | 98.7% (95.6–99.6%) | 97.0% (93.2–98.6%) | **96.7% (93.0–98.2%)** |

**Table 5. Specificity of HCV-Ab tests among HBsAg positives and negatives.** The adjusted PPV is based on an HCV prevalence of 8.5% and and HBsAg prevalence of 11%.

| Company / Product Brand | All | | HBsAg(-) | | HBsAg(-) | | Weighted |
|---|---|---|---|---|---|---|---|
| | FP/N | Specificity | FP/N | Specificity | FP/N | Specificity | Specificity |
| Abon Biopharm / OneStep HBsAg; HCV | 25/180 (13.9%) | 86.1% (80.3–90.4%) | 1/90 1.1% | 98.9% (94–99.8%) | 24/90 (26.7%) | 73.3% (63.4–81.3%) | 96.1% (90.6–97.8%) |
| CTK Biotech / OnSite HBsAg/HCVAb | 6/180 (3.3%) | 96.7% (92.9–98.5%) | 0/90 (0) | 100% (95.9–100%) | 6/90 (6.7%) | 93.3% (86.3–96.9%) | 99.3% (94.9–99.7%) |
| Cypress Diagnostics / HBV; HCV | 4/180 (2.2%) | 97.8% (94.5–99.1%) | 0/90 (0) | 100% (95.9–100%) | 4/90 (4.4%) | 95.6% (89.2–98.3%) | 99.5% (95.2–99.8%) |
| Green Gross Medical Science / Genedia HBsAg; HCV | 2/180 (1.1%) | 98.9% (96.1–99.7%) | 0/90 (0) | 100% (95.9–100%) | 2/90 (2.2%) | 97.8% (92.3–99.4%) | 99.8% (95.5–99.9%) |
| Human Diagnostic / Hexagone HBV; HCV | 4/180 (2.2%) | 97.8% (94.5–99.1%) | 2/90 (2.2%) | 97.8% (92.3–99.4%) | 2/90 (2.2%) | 97.8% (92.3–99.4%) | 97.8% (92.3–99.4%) |
| Humasis / HBsAg; HCV-Ab strip | 11/180 (6.1%) | 93.9% (89.4–96.5%) | 0/90 (0) | 100% (95.9–100%) | 11/90 (12.2%) | 87.8% (79.5–93%) | 98.7% (94.1–99.2%) |
| Intec / HBV/HCV | 4/180 (2.2%) | 97.8% (94.5–99.1%) | 0/90 (0) | 100% (95.9–100%) | 4/90 (5.4%) | 95.6% (89.2–98.3%) | 99.5% (95.2–99.8%) |
| OraSure / OraQuick HCV | 0/180 (0) | 100% (97.9–100%) | 0/90 (0) | 100% (95.9–100%) | 0/90 (0) | 100% (95.9–100%) | 100% (95.9–100%) |
| SD-Bioline / HBsAg; HBV-Ab | 2/180 (1.1%) | 98.9% (96.1–99.7%) | 0/90 (0) | 100% (95.9–100%) | 2/90 (2.2%) | 97.8% (92.3–99.4%) | 99.8% (95.5–99.9%) |
| Wondfo Biotech / OneStep HBsAg; HCV | 6/180 (3.3%) | 96.7% (92.9–98.5%) | 3/90 (3.3%) | 96.7% (90.7–98.9%) | 3/90 (3.3%) | 96.7% (90.7–98.9%) | 96.7% (90.7–98.9%) |
| **All/Average** | **64/1800 (3.6%)** | **96.4% (92.9–98.1%)** | **6/900 (0.7%)** | **99.3% (94.9–99.8%)** | **58/900 (6.4%)** | **93.6% (87.1–96.5%)** | **98.7% (94–99.4%)** |

No differences (p>0.05) in gender for either false positive or negative rates were detected for any of the tests.

## Predictive values for screening among Mongolian adults

For detection of HBsAg under the assumption of an 11% prevalence, PPV range from 73% to 100% with an average of 92%. NPV all converge to 100% in absence of FN test results. For detection of HCV-Ab under the assumption of a prevalence of 8.5%, PPV range from 40% to 100% with an average of 72%. NPV range from 99.7% to 100% (99% average).

## Discussion

Overall good results for most tests in our study demonstrates that inexpensive RDTs are highly valuable for initial HBV and HCV screening.

Detection of HBsAg using RDTs was very reliable. Of 9 kits tested, those of 6 manufacturers (ABON Biopharm, GreenCross, Humasis, InTec, SD Bioline, Wondfo) gave results in perfect agreement with the reference. The other kits gave some false positive results. However, all 90 seropositive HBsAg samples were correctly identified by all RDTs. Overall, due to their good results, Abon, GreenCross, Humasis, InTec, SD Bioline, and Wondfo RDTs can be recommended for HBV screening using blood serum.

For detection of HCV-Ab, the most striking result is the association of false positive results with HBsAg positivity of among 7 out of the 10 tests. While the cause for this effect is currently unclear, we can assume, that manufactures use similar formulations of active components in their kits. Several manufactures may have been using one component, which introduce a degree of sensitivity to HBsAg positive samples. However in previous studies it was demonstrated, that HIV positivity can have an effect on specificity of HCV-Ab RDTs [12].

Among RTDs for HCV-Ab detection, FDA-approved OraQuick demonstrated the highest diagnostic accuracy. However, this is by far the most expensive HCV-Ab test in this study [13]. Given financial constraints, a tradeoff between lower costs and slightly lower diagnostic accuracy can be advisable. Omitting HBsAg positive samples, kits by InTec, CTK and Humasis showed results in full agreement with the reference tests. However based on our results HCV-Ab test kits must be chosen with great caution especially in populations with a high prevalence of HBV.

For screening activities in Mongolia, the results highlight the main problems in practical use. Give random sampling among 1000 adults with and HCV-Ab prevalence of 8.5%, results in 85 positve and 915 negative samples. With the worst performing test with a NPV of 99.7%, we would expect 3 false negatives–potentially patients who would miss life-saving treatment due to false test results. In contrast for the same example, the lowest adjusted PPV (70.3%) would result in in 25 false positive tests which cause follow-up costs for confirmatory diagnostics.

If the tests were used to estimate the prevalence of HCV among Mongolian adults, tests from Orasure would deliver accurate of 8.5% HCV-Ab prevalence, tests from Abon would suggest a prevalence of 12.1%. Such results can lead to misguided public health interventions and in this way have a far reaching impact. Therefore, estimates of a prevalence of HBV or HCV in a population based solely on RDTs must be interpreted with a high degree of caution.

Our study was performed by well-trained laboratory personnel on serum samples. Some of the tested RDTs for HBsAg and most for HCV-Ab detection are also specified for use of whole blood. Using venipuncture or finger-stick whole blood instead of serum, could significantly simplify testing and therefore facilitate large scale testing. However, sensitivities and specificities might differ in that case and should be assessed separately.

Also it is important to point out, that for detection of HCV-Ab individuals, only patients with positive HCV-RNA test results were enrolled. Non-viremic HCV-Ab people, with possibly lower HCV-Ab concentrations (which could lead to lower RDT sensitivity values, were not included in this study.

All these issues can and should be addressed in future assessments under less defined conditions (untrained staff), simpler sample preparation (whole blood) and enrollment of non-viremic HCV-Ab positive participants, irrespective of their HCV-RNA status. Repeated assessments can also confirm a consistent quality of the manufacturers or demonstrate the opposite. For example, a study can be conducted at local primary care centers without any training for the care workers, other than the information sheet of each manufacturer.

In conclusion, our prospective study demonstrates that inexpensive RDTs can provide a good alternative for screening for HBV and/or HCV. This is especially true in scenarios in low-income countries, where the alternative would mean no screening at all. For population-wide screening for HBV and HCV e.g. within the Screening Campaign of the Hepatitis Prevention, Control, and Elimination Program in Mongolia, point-of-care screening using RDTs is an option that can be recommended.

## Author Contributions

**Conceptualization:** Zulkhuu Genden, Dahgwahdorj Yagaanbuyant, Naranbaatar Dashdorj, Naranjargal Dashdorj.

**Data curation:** Ganbolor Jargalsaikhan, Delgerbat Boldbaatar.

**Formal analysis:** Miriam Eichner, Andreas Bungert.

**Funding acquisition:** Naranbaatar Dashdorj.

**Investigation:** Delgerbat Boldbaatar, Purevjargal Bat-Ulzii, Oyungerel Lkhagva-Ochir, Bekhbold Dashtseren, Erdenebayar Namjil, Alimaa Tuya, Naran Gurjav, Altankhuu Mordorj, Naranjargal Dashdorj.

**Methodology:** Miriam Eichner, Odgerel Oidovsambuu, Andreas Bungert, Naranbaatar Dashdorj, Naranjargal Dashdorj.

**Project administration:** Ganbolor Jargalsaikhan, Naranbaatar Dashdorj, Naranjargal Dashdorj.

**Resources:** Erdenebayar Namjil, Alimaa Tuya, Naran Gurjav, Naranbaatar Dashdorj.

**Supervision:** Ganbolor Jargalsaikhan, Zulkhuu Genden, Dahgwahdorj Yagaanbuyant, Altankhuu Mordorj, Andreas Bungert, Naranbaatar Dashdorj, Naranjargal Dashdorj.

**Validation:** Andreas Bungert.

**Visualization:** Andreas Bungert.

**Writing – original draft:** Ganbolor Jargalsaikhan, Miriam Eichner.

**Writing – review & editing:** Miriam Eichner, Andreas Bungert, Naranbaatar Dashdorj, Naranjargal Dashdorj.

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
