## [Decision Letter · Decision Letter 0]

11 Feb 2020

PONE-D-19-25400

Performance of Commercially Available Rapid Diagnostic Tests for Viral Hepatitis B and C Screening in Serum Samples

PLOS ONE

Dear Dr Bungert,

Thank you for submitting your manuscript to PLOS ONE. After careful consideration, we feel that it has merit but does not fully meet PLOS ONE’s publication criteria as it currently stands. Therefore, we invite you to submit a revised version of the manuscript that addresses the points raised during the review process both on analysis of the results, methodology, discussion and references.

We would appreciate receiving your revised manuscript by 2 months due. To enhance the reproducibility of your results, we recommend that if applicable you deposit your laboratory protocols in protocols.io, where a protocol can be assigned its own identifier (DOI) such that it can be cited independently in the future. For instructions see: http://journals.plos.org/plosone/s/submission-guidelines#loc-laboratory-protocols

We look forward to receiving your revised manuscript.

Kind regards,

Isabelle Chemin, PhD

Academic Editor

PLOS ONE

Journal Requirements:

Reviewers' comments:

Reviewer's Responses to Questions

**Comments to the Author**

1. Is the manuscript technically sound, and do the data support the conclusions?

Reviewer #1: Partly

2. Has the statistical analysis been performed appropriately and rigorously? 

Reviewer #1: No

3. Have the authors made all data underlying the findings in their manuscript fully available?

Reviewer #1: No

4. Is the manuscript presented in an intelligible fashion and written in standard English?

Reviewer #1: No

5. Review Comments to the Author

Reviewer #1: This work shows the results of the comparison of a comprehensive number of rapid diagnostic tests (RDT) for hepatitis C antibodies on a relatively modest number of samples previously characterized by ELISA (used as gold-standard).

MAJOR POINTS

1. L76. In the inclusion criteria, were patients with previous/current antiviral treatment included or excluded. If included, please specify treatment details.

2. The authors use routine serology as the gold standard to evaluate the performance of RTDs. Please provide methods and kit details of the standard assays used for HBsAg and HCV-Ab testing (e.g. ELISA, MEIA, kit name, manufacturer, etc.)

3. For the statistical analysis, the authors used Wilson score for assessing the specifity and sensitivity of the RDTS, and this reviewer interprets that likelihood ratios were also calculated. This is not clarified in the methods, and reference 7 on likelihood tests is not referenced in the main text.

4. In this line, the authors somewhat interchange concepts that are in fact different variables: test performance, sensitivity, and diagnostic accuracy. The performance is the evaluation of all variables (true positives, negatives, sensitivity, specificity, negative and positive predictive values, likelihood ratios for positive and negative results and diagnostic accuracy). Likelihood ratios and diagnostic accuracy values are not provided (see below).

5. L126 and Table 2. Can the authors provide more data on the studied cohorts, such as ALT, AST, GGT, fibrosis scores/stage of liver disease (if available), other HBV serological markers (e.g. HBeAg), risk factors, etc.?

6. L139-143. Here there is a confusion between performance, sensitivity and accuracy (see above). Please, use appropriate terms.

7. Table 3. I would suggest ordering the tests by sensitivity and then by specificity. Add percentages for quotients. Add columns with values for positive and negative predictive values, likelihood ratios and calculated diagnostic accuracy (including 95% CI). See examples:. Larrat, et al J. Clin. Virol. 55(3) (2012) 220–225; Cloherty, et al. J. Clin.Microbiol. 54 (2016) 265–273.

8. The same applies for Table 4.

9. L158. “…in HCV-Ab negative, HBsAg positive sera than in sera from healthy…” Do the authors have any explanation for this? Include in the discussion.

10. L160. I believe that figure 1 is redundant with the tables. The authors may want to transfer details provided in the caption to the text, or to a new table describing the details for all the discrepant results.

11. In the discussion the authors argue about tests performance, the value of diagnostic accuracy, when this was not calculated, etc-…please, re-write the discussion section completely after the lacking variables have been analyzed

12. L203-205. Delete paragraph

13. L206-209. Rephrase, focus on the importance of tasting in the field with real world conditions on your environment, and what how to identify HCV-Ab positives by RDTs with negative HCV-RNA.

14. The list of references is quite scarce, there have been a quite high number of paper published in the topic. For a reference see for instance Peeling et al. BMC Infect Dis. 2017 Nov 1;17(Suppl 1):699

MINOR POINTS

L43 change “less good” for, e.g. “somewhat lower”

L55. “…are diagnosed early.2

L63: “Therefore, the performance….”

L65. “The Aim of…”

L 74. “…from the screening registry…”

L79. “Inclusion criteria”

L89. “venous blood”

L92. “blood drawn”

L 96 quantitative RT-PCR? If so, for quantitation of HBV-DNA and HCV-RNA levels?

Table 1, Orasure: delete (!) sign on distributor details

L130. “…from the HBsAg positive group, HBV-DNA was negative.

L131. “...from the HCV-Ab positive group, HCV-RNA was positive but below the limit of quantitation of the assay (specify LOQ)

L149 “…the OraQuick HCV Ab test…”

L150. “…for all HCV-Ab…”

L151: “…and average specificity was….”

L200. “…HCV-Ab indivduals…”

L201. “…lower HCV-Ab levels (which could lead to a lower RTD senstitivity values) were not included in this study.”

6. PLOS authors have the option to publish the peer review history of their article (what does this mean?). If published, this will include your full peer review and any attached files.

Reviewer #1: No

---

## [Author Response · Author response to Decision Letter 0]

7 Apr 2020

Response to reviewers has been attached as a file.

---

## [Decision Letter · Decision Letter 1]

9 Jun 2020

Sensitivity and Specificity of Commercially Available Rapid Diagnostic Tests for Viral Hepatitis B and C Screening in Serum Samples

PONE-D-19-25400R1

Dear Dr. Bungert,

We are pleased to inform you that your manuscript has been judged scientifically suitable for publication and will be formally accepted for publication once it complies with all outstanding technical requirements.

With kind regards,

Isabelle Chemin, PhD

Academic Editor

PLOS ONE

Additional Editor Comments (optional):

Reviewers' comments:

Reviewer's Responses to Questions

**Comments to the Author**

1. If the authors have adequately addressed your comments raised in a previous round of review and you feel that this manuscript is now acceptable for publication, you may indicate that here to bypass the “Comments to the Author” section, enter your conflict of interest statement in the “Confidential to Editor” section, and submit your "Accept" recommendation.

Reviewer #1: All comments have been addressed

2. Is the manuscript technically sound, and do the data support the conclusions?

Reviewer #1: Yes

3. Has the statistical analysis been performed appropriately and rigorously? 

Reviewer #1: Yes

4. Have the authors made all data underlying the findings in their manuscript fully available?

Reviewer #1: Yes

5. Is the manuscript presented in an intelligible fashion and written in standard English?

Reviewer #1: Yes

6. Review Comments to the Author

Reviewer #1: The authors have addressed the questions raised, and they only need to coorect some minor typological errors / word repetitions in the text.

7. PLOS authors have the option to publish the peer review history of their article (what does this mean?). If published, this will include your full peer review and any attached files.

Reviewer #1: No

---

## [Editor Report · Acceptance letter]

6 Jul 2020

PONE-D-19-25400R1 

Sensitivity and Specificity of Commercially Available Rapid Diagnostic Tests for Viral Hepatitis B and C Screening in Serum Samples 

Dear Dr. Bungert:

I'm pleased to inform you that your manuscript has been deemed suitable for publication in PLOS ONE. Congratulations! Your manuscript is now with our production department. 

Kind regards, 

on behalf of

Mrs Isabelle Chemin 

Academic Editor

PLOS ONE